# Is Tocilizumab Plus Dexamethasone Associated with Superinfection in Critically Ill COVID-19 Patients?

**DOI:** 10.3390/jcm11195559

**Published:** 2022-09-22

**Authors:** Fabrice Camou, Nahéma Issa, Mojgan Hessamfar, Olivier Guisset, Gaëlle Mourissoux, Stéphane Pedeboscq, Aimée Minot, Fabrice Bonnet

**Affiliations:** 1CHU BORDEAUX, Intensive Care and Infectious Diseases Department, Saint-Andre Hospital, 33075 Bordeaux, France; 2CHU BORDEAUX, Internal Medicine and Infectious Diseases Department, Saint-Andre Hospital, 33075 Bordeaux, France; 3CHU BORDEAUX, Pharmacy Department, Saint-Andre Hospital, 33075 Bordeaux, France; 4Bordeaux Population Health, INSERM U1219, University of Bordeaux, 33000 Bordeaux, France

**Keywords:** COVID-19, dexamethasone, tocilizumab, superinfection

## Abstract

Background: Dexamethasone and tocilizumab are used to treat severely ill COVID-19 patients admitted to intensive care units (ICUs). We explored whether combination therapy increased the risk of superinfection compared to dexamethasone alone. Methods: This observational, retrospective study included critically ill COVID-19 adult patients admitted to our ICU because of respiratory failure. Patients received dexamethasone with (Group 1) or without (Group 2) tocilizumab. Data were collected from electronic medical files. Results: A total of 246 patients were included, of whom 150 received dexamethasone and tocilizumab, while 96 received dexamethasone alone. Acute respiratory distress syndrome was evident on admission in 226 patients, 56 of whom required mechanical ventilation (MV). Superinfections, mainly respiratory, were diagnosed in 59 patients, including 34/150 (23%) in Group 1 and 25/96 (26%) in Group 2 (*p* = 0.32). After multivariate analysis, the factors associated with a higher risk of superinfection included hematological malignancy (hazard ratio (HR): 2.47 (1.11–5.47), *p* = 0.03), MV (HR: 3.74 (1.92–7.26), *p* = 0.0001), and a higher SAPS-II score on admission (HR: 1.03 (1.01–1.06), *p* = 0.006). Conclusion: In critically ill COVID-19 patients, the addition of tocilizumab to dexamethasone was not associated with an increased risk of superinfection.

## 1. Introduction

Since July 2020, most COVID-19 patients admitted to an intensive care unit (ICU) have been given dexamethasone, while some have also received tocilizumab, in line with the guidelines [1,2]. Data on the risk of superinfection following implementation of these therapeutic strategies remain scarce in ICUs, and it is not clear if there is an increased risk of superinfection in patients treated with these two anti-inflammatory drugs (tocilizumab plus dexamethasone) [3,4,5,6].

The main objective of our study was to determine the incidence of superinfections associated with COVID-19 patients in an ICU. The secondary aims were to characterize the superinfections and calculate the length of ICU stay and 28-day mortality rate.

## 2. Methods

### 2.1. Study Design and Participants

This retrospective, observational, single-center study was conducted in Bordeaux University Hospital between March 2020 and August 2021. The study population included all adult patients diagnosed with COVID-19 by RT-PCR on nasopharyngeal swabs admitted to the ICU because of respiratory failure, requiring oxygen at >4 L/min; all patients were given dexamethasone. We divided the study population into two groups based on whether they did (Group 1) or did not (Group 2) receive additional tocilizumab within 24 h of admission. 

### 2.2. Data Collection

Data were collected from electronic medical files using DxCare software (Le Plessis Robinson, Dedalus France, version 779). Demographic data, comorbidities (hypertension, obesity (body mass index > 30 kg/m^2^), diabetes, malignancy, and immunodeficiency), and clinical, microbiological, and radiological parameters were recorded. Immunocompromised patients included patients with malignancy or non-malignant immunodeficiency (HIV infection, systemic disease treated with long-term steroids (>30 days) or any immunosuppressive drug, or solid-organ transplantation). On admission, viral and bacterial infections were systematically screened using pharyngeal multiplex PCR, urinary antigen tests, blood cultures, and sputum cultures (where possible). During hospitalization, if clinical deterioration was evident, blood cultures, urine cultures, and tracheal aspiration or bronchoalveolar lavage were performed.

### 2.3. Definitions and Management of Infections

Superinfection (secondary infection) was defined as a clinical infection associated with a positive pathogenic bacterial or fungal test > 3 days after hospital admission. Coinfection was defined as an infection diagnosed 0–2 days after admission. Fungal superinfections were systematically screened by serum galactomannan and *Aspergillus* spp. PCR every week. On admission, all patients were given ceftriaxone (2 g/day) for 5 days and dexamethasone (6 mg/day) for 10 days. Those with a fever associated with increased inflammatory parameters (C-reactive protein > 75 mg/L) were also given one dose of tocilizumab (8 mg/kg) within the first 24 h, according to the local guidelines and our infectious disease protocols. 

As this retrospective study used anonymized healthcare data, patient consent was not required by French legislation. In line with French law and the rules of the French Data Protection Authority, the handling of these data for retrospective research purposes was declared to the Data Protection Officer of the University Hospital of Bordeaux, who acknowledged the use of anonymized data. Patients were notified about the use of their anonymized data via a departmental booklet.

### 2.4. Statistical Analysis

Continuous variables are described as the median and interquartile range (IQR), while categorical variables are expressed as frequencies (%). Group comparisons of continuous variables were performed using Student’s *t*-test. Categorical data were compared using the χ^2^ test for count data. To analyze the risk of superinfection, survival analysis was performed using a multivariate Cox model, via a stepwise regression with a forward selection method to identify the parameters most associated with superinfection. The *p*-value cut-off for inclusion in the multivariate analysis was <0.20 in order to include the relevant baseline and demographic variables.

## 3. Results

### 3.1. Patient Characteristics

During the study period, 246 patients (72% males) with severe COVID-19 were enrolled, of whom 150 received dexamethasone and tocilizumab (Group 1) and 96 received dexamethasone without tocilizumab (Group 2) (Table 1). The median period from symptom onset to ICU admission was 8 days in both groups. Overall, 26% of patients exhibited malignant or non-malignant immunodeficiency (22% in Group 1 vs. 32% in Group 2, *p* = 0.07) and 23% had no chronic comorbidity (26% vs. 19%, *p* = 0.19). The proportion of males was 79% in Group 1 and 62% in Group 2 (*p* = 0.005). The median SAPS-II and SOFA scores were 23 and 3, respectively, with no difference between the two groups. Acute respiratory distress syndrome (ARDS, PaO_2_/FiO_2_ ≤ 300 mmHg) was present on admission in 226 patients. Severe ARDS (PaO_2_/FiO_2_ ≤ 100 mmHg) was diagnosed in 35% of the patients in Group 1 and 16% of those in Group 2 (*p* < 0.01). Fifty-six patients required mechanical ventilation (MV): 25% of those in Group 1 and 20% of those in Group 2 (*p* = 0.23). Ninety-nine patients needed non-invasive ventilation (45% of those in Group 1 and 22% of those in Group 2, *p* < 0.001). The C-reactive protein and ferritin levels were significantly higher in Group 1 than Group 2. On admission, 33 patients had a bacterial respiratory coinfection, with no difference between the groups.

### 3.2. Outcomes

During the study period, 59 patients (24%) developed 66 documented superinfections (34/150 (23%) in Group 1 and 25/96 (26%) in Group 2, *p* = 0.32) (Table 2). The superinfections were healthcare- or ventilator-associated pneumonia (H/VAP) in 39 patients, with Gram-negative bacillus documented in 19 cases (45%). These superinfections occurred in a median of 10 days after admission. There were 11 cases of COVID-associated pulmonary aspergillosis (CAPA), which developed after a median of 9 days after admission. Eight patients had bacteremia (including three with catheter-related bacteremias), and six had urinary tract infections. One patient developed *Pneumocystis jirovecii* pneumonia and one had a *Clostridioides difficile* infection.

The median ICU stay was 19 days in the case of superinfection vs. 5 days in patients without superinfection (*p* < 0.01). The 28-day mortality rate of patients with superinfection was 36% (21/59), compared with 7% in the others (14/187) (*p* < 0.01). The 28-day mortality was 13% in Group 1 and 17% in Group 2 (*p* = 0.38). After multivariate analysis, factors associated with a higher risk of superinfection included hematological malignancy (hazard ratio (HR): 2.47; 95% confidence interval (CI): 1.11–5.47, *p* = 0.03), invasive MV (HR: 3.74; 95% CI: 1.92–7.26, *p* = 0.0001), and the SAPS-II score on admission (HR: 1.03; 95% CI: 1.01–1.06; *p* = 0.006) (Table 3). The use of tocilizumab in addition to dexamethasone was not associated with an increased risk of superinfection (HR: 0.61; 95% CI: 0.33–1.06; *p* = 0.11). Neither gender nor obesity was associated with superinfection.

## 4. Discussion

In this retrospective, observational study, 24% of the patients admitted to the ICU for severe COVID-19 who received dexamethasone developed a superinfection. Early studies of COVID-19 patients reported lower rates of superinfection. At that time, most patients with severe COVID-19 received empirical broad-spectrum antimicrobials, influenced by our experience with influenza and the risk of co-infection. Empirical antimicrobial therapy is no longer used systematically, to avoid antibiotic-resistant superinfections [2]. Beginning in the summer of 2020, randomized trials have shown that dexamethasone improved the survival of COVID-19 patients and the use thereof became the standard of care in ICUs [7]. More recently, IL-6 receptor blockers have been recommended for severe COVID-19 when there is uncertainty about adverse infections, especially in the ICU [1,2,8].

Following such changes in therapeutic strategies, the risk of superinfection became a potential issue, particularly in patients receiving combination therapy. Other ICU studies reported harmful effects of dexamethasone; the rates of superinfection ranged from 27% to 54% [3,5,6,9]. In the ESICM UNITE-COVID study, after adjusting the data of a propensity-matched cohort for potential confounders, 71% of patients receiving corticosteroids developed ICU-acquired infections (ICU-AI) compared with 52% of those not prescribed such drugs [6]. In an observational Norwegian study, dexamethasone use was strongly and independently associated with superinfection in COVID-19 patients on invasive MV (HR: 3.7; 95% CI: 1.80–7.61; *p* < 0.001) [5]. Data on the impact of tocilizumab (in addition to dexamethasone) on the risk of superinfection in the context of COVID-19 are limited and conflicting. 

In our study, we found no increased risk of superinfection in patients treated with a combination of tocilizumab and dexamethasone (23%) compared to dexamethasone alone (26%), although the former patients exhibited more inflammation on admission, required more high-flow oxygen, and experienced longer ICU stays. These results are in line with those of the observational ESICM UNITE-COVID study, where the use of tocilizumab in 132 patients was not associated with an increased risk of ICU-AI: 46% of patients receiving tocilizumab developed infections compared to 56% of those not on the drug (*p* = 0.04) [6]. The global risk of ICU-AI was twice higher than in our study to develop an ICU-AI. In contrast, albeit outside of the ICU context, tocilizumab was associated with an increased risk of coinfection in patients with cancer [10].

The proportion of patients requiring MV was 23% in our study; bacterial H/VAP was the main cause of superinfection, affecting 24% of the patients. In comparison, in the ESICM UNITE-COVID study, 83% of patients required MV and the H/VAP rate was 77%. Saade et al. reported a 38% VAP rate in their patient population, 54% of whom were on MV [6,9]. The lower respiratory superinfection rate of our study is partly explained by the lower MV rate, which is the main contributor to superinfection [9]. 

Of the various respiratory superinfections, CAPA was diagnosed in 4% of patients; there was no difference in the diagnostic rate between the two groups, and this proportion is in agreement with other studies [5,6]. In the MYCOVID study, Gangneux et al. showed that the combination of anti-IL6 and dexamethasone was associated with a significantly increased risk of CAPA (HR: 2.71; 95% CI: 1.12–6.56; *p* = 0.027) in patients on MV [11]. In that study, respiratory fungal microorganisms were analyzed during MV. In our study, the lower CAPA rate is explained by the lower MV rate, which is the main driver of fungal infection. 

The main factors associated with the risk of superinfection in our study were invasive MV, the SAPS-II score on admission, and hematological malignancy, rather than adjunctive therapy with tocilizumab. In the Norwegian study, superinfection was independently associated with pre-existing autoimmune disease and the length of ICU stay, but only 4.5% of patients had malignancies compared with 18% in the present study [5].

Our results may be explained by multiple factors. The ICU was in a non-inundated area of France with a low prevalence of extended-spectrum beta-lactamase. Systematic screening of coinfections and superinfections was protocolized with a multidisciplinary approach, and the experience from other centers prompted us to delay MV when possible. Being a retrospective, single-center design is the main limitation of this study, and must be kept in mind when interpreting the results. 

## 5. Conclusions

Tocilizumab in combination with dexamethasone is not associated with an increased risk of superinfection in critically ill COVID-19 patients. Factors associated with superinfection included hematological malignancy, MV, and the SAPS-II score on admission. 

## Figures and Tables

**Table 1 jcm-11-05559-t001:** Comparison of ICU-hospitalized COVID-19 patients according to immunomodulatory treatment on admission.

Variable		All Patients *n* = 246	Dexamethasone with Tocilizumab (Group 1) *n* = 150	Dexamethasone alone (Group 2) *n* = 96	*p* Value
Male sex, *n* (%)		178 (72)	118 (79)	60 (62)	0.005
Age, years, median (IQR)		61 (50–71)	60 (49–70)	62 (53–72)	0.20
Age > 65 years, *n* (%)		101 (41)	56 (37)	45 (47)	0.09
SAPS-II on admission, median (IQR)		27 (22–36)	27 (22–36)	27 (22–36)	0.40
SOFA score on admission, median (IQR)		3 (3–4)	3 (3–4)	3 (2–4)	0.08
Charlson Comorbidity Index, median (IQR)		3 (1–5)	2 (1–4)	3 (1–5)	0.05
Concomitant bacterial infection, *n* (%)		33 (13)	19 (13)	14 (15)	0.40
Coexisting comorbidities, *n* (%)	Body Mass Index > 30 kg/m^2^	86 (35)	53 (35)	33 (34)	0.49
	Solid cancer	24 (10)	13 (9)	11 (11)	0.31
	Hematologic malignancy	21 (8)	11 (7)	10 (10)	0.27
	Diabetes mellitus	49 (20)	30 (20)	19 (20)	0.55
	Arterial hypertension	88 (36)	55 (37)	33 (34)	0.41
	Non-malignant immunodeficiency	24 (10)	11 (7)	13 (13)	0.08
ARDS (PaO_2_/FiO_2_ ≤ 300 mmHg), *n* (%)		226 (92)	141 (94)	85 (89)	0.13
CRP on admission, mg/L, median (IQR)		140 (84–219)	166 (102–233)	103 (60–154)	<0.001
Ferritin on admission, ng/mL, median (IQR)		1536 (649–2624)	1797 (1114–2907)	845 (453–2043)	<0.001
Invasive mechanical ventilation, *n* (%)		56 (23)	37 (25)	19 (20)	0.23
Non-invasive ventilation, *n* (%)		76 (31)	53 (35)	23 (24)	0.06
High-flow oxygen, *n* (%)		177 (72)	118 (79)	59 (61)	0.003
Length of mechanical ventilation, days, median (IQR)		14 (9–28)	14 (9–28)	15 (7–27)	0.89
Prone position, *n* (%)		23 (9)	20 (13)	3 (3)	0.007
Vasopressor support, *n* (%)		19 (8)	11 (7)	8 (8)	0.77
Renal replacement therapy, *n* (%)		5 (2)	4 (3)	1 (1)	0.68
Superinfection, *n* (%)		59 (24)	34 (23)	25 (26)	0.32
COVID-associated pulmonary aspergillosis, *n* (%)		11 (4)	6 (4)	5 (5)	0.44
Length of ICU stay, days, median (IQR)		6 (3–14)	7 (4–17)	4 (2–8)	<0.001
28-day mortality, *n* (%)		22 (9)	10 (7)	12 (12)	0.09
60-day mortality, *n* (%)		33 (13)	17 (11)	16 (17)	0.16

**Table 2 jcm-11-05559-t002:** Microbial etiology of 66 episodes of superinfections in 246 hospitalized patients with COVID-19.

	Dexamethasone with Tocilizumab (Group 1) *n* = 150				Dexamethasone Alone (Group 2) *n* = 96		
	Respiratory Sample	Blood	Urine	Stools	Respiratory Sample	Blood	Urine
**Gram-negative bacteria (GNB)**							
Enterobacterales							
*Klebsiella pneumoniae*	3						1
*Escherichia coli*		1	1			1	1
*Enterobacter* spp.	2				2		
*Citrobacter* spp.	2						
*Serratia* spp.	1						
Non-fermenting GNB							
*Pseudomonas aeruginosa*	2	1			2		
*Acinetobacter* spp.	2					1	
*Stenotrophomonas maltophilia*	1				1		
**Gram-positive bacteria**							
*Enterococcus* spp.	5		1		1	1	2
*Staphylococcus aureus*	3				2	1	
*Streptococcus* spp.	6				4		
*Coagulase negative staphylococci*		2					
*Clostridium difficile*				1			
**Fungi**							
*Aspergillus* spp.	5				6		
*Pneumocystis jiroveci*					1		

**Table 3 jcm-11-05559-t003:** Univariate and multivariate analysis of predictors independently associated with superinfection in 246 hospitalized patients with COVID-19.

Predictor	*n* (%)	Univariate Analysis		Multivariate Analysis	
		HR	*p* Value	HR (95% CI)	*p* Value
Male sex	178 (72)	1.16	0.69		
Age > 65 years	101 (41)	2.02	0.02	1.73 (0.94–3.17)	0.08
Body Mass Index > 30 kg/m^2^	86 (35)	0.62	0.15	1.08 (0.53–2.20)	0.84
Solid cancer	24 (10)	1.29	0.56		
Hematologic malignancy	21 (8)	2.74	0.01	2.47 (1.11–5.47)	0.03
Diabete mellitus	49 (20)	1.27	0.47		
Arterial hypertension	88 (36)	0.99	0.98		
Non-malignant immunodeficiency	24 (10)	1.31	0.61		
Invasive mechanical ventilation	56 (23)	4.31	10^−4^	3.74 (1.92–7.26)	0.0001
Concomitant bacterial infection	33 (13)	2.04	0.05	1.35 (0.59–3.14)	0.48
Tocilizumab	150 (61)	0.47	0.01	0.61 (0.33–1.13)	0.11
SAPS-II on admission	246 (100)	1.07	10^−4^	1.03 (1.01–1.06	0.006
CRP on admission	246 (100)	1.00	0.45		
Ferritin on admission	246 (100)	1.00	0.63		

## Data Availability

The data sets used during the study are available from the corresponding author on reasonable request.

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
