# Peer review of "Is Tocilizumab Plus Dexamethasone Associated with Superinfection in Critically Ill COVID-19 Patients?"

_jcm, 2022, doi:10.3390/jcm11195559_

Round 1
Reviewer 1 Report
In this observational retrospective single center study with the aim to assess if combination therapy with dexamethasone and tocilizumab was associated with an increased risk of superinfection versus dexamethasone alone, a total of 246 adult critically ill COVID-19 patients were included, 150 receiving dexamethasone and tocilizumab and 96 dexamethasone alone. No significant difference was found in incidence of superinfections between the two groups and, in multivariate analysis, significant risk factors for superinfections included hematologic malignancy, MV, and SAPS-II score on admission. The aim and design of the present study are interesting.
However, the following major issues should be addressed before the publication.
11. I have some important concerns about how the statistical analysis was conducted. In fact, in Method section, authors state that “to analyze the risk of superinfection, survival analysis was performed using a multivariate Cox model”. Instead, results of univariate and multivariate analyses are expressed as OR (95% CI). In addition, in multivariate analysis, ORs (95% CI) were reported for other variables, even some which resulted not significant (please note that the P value cut-off for inclusion in multivariate analysis was not defined in methods: was it ≤0.15?) in univariate analysis: was the multivariate model adjusted for all these variables? If yes, please specify it and the reasons in the Methods section. Please provide a careful and correct revision of the statistical analysis.
22. Table 2: please add numbers and percentages of patients for each variable included.
33. In the Methods section authors state that “a superinfection, synonymous with secondary infection, was defined as a clinical infection associated with a positive specimen with pathogenic bacterial or fungal microorganism obtained > 3 days after hospital admission”. According to this definition, each episode of superinfection was microbiologically diagnosed. However, this is not clear in the text. I suggest to adding a new table with a detailed description of superinfections according to clinical site and microbiological aetiology.
Author Response
Dear reviewer, Thank you for your interest in our work. We carefully read the comments and replied to them point by point. We modified the manuscript to incorporate their feedback and improve the quality of our report. As requested, a final version with the highlighted changes (in red) is attached.
We hope that our responses to the comments and the modifications provided in the manuscript have met your expectations.
Best regards,
Dr Fabrice CAMOU for all co authors
Reviewer #1:
- I have some important concerns about how the statistical analysis was conducted. In fact, in Method section, authors state that “to analyze the risk of superinfection, survival analysis was performed using a multivariate Cox model”. Instead, results of univariate and multivariate analyses are expressed as OR (95% CI). In addition, in multivariate analysis, ORs (95% CI) were reported for other variables, even some which resulted not significant (please note that the P value cut-off for inclusion in multivariate analysis was not defined in methods: was it ≤0.15?) in univariate analysis: was the multivariate model adjusted for all these variables? If yes, please specify it and the reasons in the Methods section. Please provide a careful and correct revision of the statistical analysis.
Thank you for this comment. We have corrected the initials in our tables and replaced OR by HR (Hazard Ratio).The p value cut-off for inclusion in multivariate analysis was ≤ 0.20.
- Table 2: please add numbers and percentages of patients for each variable included.
Table 2 renamed table 3 has been modified according to this comment.
- In the Methods section authors state that “a superinfection, synonymous with secondary infection, was defined as a clinical infection associated with a positive specimen with pathogenic bacterial or fungal microorganism obtained > 3 days after hospital admission”. According to this definition, each episode of superinfection was microbiologically However, this is not clear in the text. I suggest to adding a new table with a detailed description of superinfections according to clinical site and microbiological aetiology.
We added a third table describing microbiological data as requested
Reviewer 2 Report
The publication titled: Are tocilizumab plus dexamethasone associated with superinfection in critically ill COVID-19 patients?
J. Clin. Med. 2021, 10, x. https://doi.org/10.3390/xxxxx www.mdpi.com/journal/jcm
In the manuscript, the authors intend to assess if specific therapy (GCS + TCZ, n=150 – group 1 vs. GCS, n=96, group 2) was associated with an increased risk of superinfection. This retrospective study analyzed critically ill COVID-19 patients with respiratory failure (mainly with ARDS, n=246). Superinfections, primarily respiratory, were diagnosed in 23% in group 1 and 26% in group 2, and the differences were not statistically significant. In the multivariate analysis, factors associated with a higher risk of superinfection included hematologic malignancy, mechanical ventilation, and SAPS-II score on admission. The authors concluded that critically ill COVID-19 patients treated with combined therapy TCZ and GCS did not present an increased risk of superinfection. The submitted data are interesting; however, a few questions are raised:
1. In the study, the male gender predominated. Does it mean more severe symptoms are observed in males than females? Have you seen the differences between males and females in such values as CRP, SAPS II, Charlson Comorbidity Index, Concomitant bacterial infection, and BMI?
2. Since obesity is one factor that influences the complications and treatment effects of COVID-19 infections, many authors describe more severe complications during SARS-CoV2 infection. It would be interesting if the authors analyzed the differences between obese and normal-weight patients. Did you observe the differences between CRP, SAPS II, Charlson Comorbidity Index, Concomitant bacterial infection, and other parameters?
Author Response
-
Dear reviewer,
Thank you for your interest in our work. We carefully read the comments and replied to them point by point. We modified the manuscript to incorporate their feedback and improve the quality of our report. As requested, a final version with the highlighted changes (in red) is attached.
We hope that our responses to the comments and the modifications provided in the manuscript have met your expectations.
Best regards,
Dr Fabrice CAMOU for all co authors
In the study, the male gender predominated. Does it mean more severe symptoms are observed in males than females? Have you seen the differences between males and females in such values as CRP, SAPS II, Charlson Comorbidity Index, Concomitant bacterial infection, and BMI?
Thank you for this comment. In the literature concerning severe COVID-19, the proportion of male is often higher and much higher than female. For example, in the French COVID-ICU cohort, 74% of severe COVID-19 patients were male [1]. We added a sentence in the manuscript about these results line 130.
Table a. Comparative data between males and females
|
|
Males n=178 |
Females N=68 |
p-value |
|
Mean CRP, mg/L (SD) |
167 (94) |
124 (78) |
0.001 |
|
Mean SAPS II (SD) |
30 (11) |
28 (13) |
0.052 |
|
Mean Charlson Comorbidity index (SD) |
3 (3) |
3 (3) |
0.410 |
|
Concomitant bacterial infection, n (%) |
22 (12) |
11 (16) |
0.277 |
|
BMI > 30 kg/m2, n (%) |
54 (30) |
32 (47) |
0.011 |
- Since obesity is one factor that influences the complications and treatment effects of COVID-19 infections, many authors describe more severe complications during SARS-CoV2 infection. It would be interesting if the authors analyzed the differences between obese and normal-weight patients. Did you observe the differences between CRP, SAPS II, Charlson Comorbidity Index, Concomitant bacterial infection, and other parameters?
In obese patients, mean CRP, SAPS II and Charlson comorbidity index were lower than in non-obese patients. This could be explained by our specific population with a high proportion of immunocompromised patients. Only two of them (3%) are obese in comparison with 84 (34%) in the non-immunodeficiency population. This point has not been developed in our manuscript due to the format of the article (brief report).
Table b. Comparative data between obese (BMI > 30 kg/m2) and non-obese patients
|
|
Obese patients n=86 |
Non-obese patients N=160 |
p-value |
|
Mean CRP, mg/L (SD) |
137 (82) |
165 (96) |
0.017 |
|
Mean SAPS II (SD) |
26 (10) |
31 (12) |
< 0.001 |
|
Mean Charlson Comorbidity index (SD) |
2 (2) |
4 (3) |
< 0.001 |
|
Concomitant bacterial infection, n (%) |
10 (12) |
23 (14) |
0.347 |
Reviewer 3 Report
This manuscript aimed to determine whether the combination therapy of Dexamethasone and tocilizumab increased the risk of superinfection compared to Dexamethasone alone in COVID-19 patients. 246 patients were included in this study, of whom 150 received Dexamethasone and tocilizumab and 96 received Dexamethasone without tocilizumab. The results of the study showed that the combination of dexamethasone and tocilizumab was not associated with an increased risk of superinfection. However, there are some problems as follows:
1. The writing of the article was not rigorous enough, such as the part of speech of “microorganism" in line 88 should be consistent with “bacterial” and “fungal” in line 87.
2. There are some grammatical errors in the manuscript, such as “are” in the title should be “is”, “increase” in line 173 should be “increased”, “a ICU-AI” in line 180 should be “an ICU-AI”, and so on.
3. It is suggested to check the full text and modify the language to improve the quality of the manuscript.
4. It is suggested to add a lower border line to beautify the table.
Author Response
Dear reviewer
Thank you for your interest in our work. We carefully read the comments and replied to them point by point. We modified the manuscript to incorporate their feedback and improve the quality of our report. As requested, a final version with the highlighted changes (in red) is attached.
We hope that our responses to the comments and the modifications provided in the manuscript have met your expectations.
Best regards,
Dr Fabrice CAMOU for all co authors
1-The writing of the article was not rigorous enough, such as the part of speech of “microorganism" in line 88 should be consistent with “bacterial” and “fungal” in line 87.
The sentence has been modified.
2-There are some grammatical errors in the manuscript, such as “are” in the title should be “is”, “increase” in line 173 should be “increased”, “a ICU-AI” in line 180 should be “an ICU-AI”, and so on.
The errors have been corrected in the manuscript.
3-It is suggested to check the full text and modify the language to improve the quality of the manuscript.
Thank you for your suggestion: the full text has been checked by an English native speaker.
- It is suggested to add a lower border line to beautify the table.
A lower border line was added in the tables.
We added a third table describing microbiological data as requested